# Comparison between Bilateral Ultrasound-Guided Quadratus Lumborum Block and Sacrococcygeal Epidural in Cats Undergoing Ovariectomy

**DOI:** 10.3390/vetsci11010025

**Published:** 2024-01-08

**Authors:** José Diogo dos-Santos, Mário Ginja, João Martins, Patrícia Cabral, Sofia Alves-Pimenta, Lénio Ribeiro, Pablo E. Otero, Bruno Colaço

**Affiliations:** 1VetOeiras—Veterinary Hospital, 2780-114 Oeiras, Portugal; 2Department of Veterinary Science, University Lusófona, 1749-024 Lisbon, Portugal; 3CECAV—Animal and Veterinary Research Centre UTAD, University of Trás-os-Montes and Alto Douro, 5000-801 Vila Real, Portugal; 4Associate Laboratory for Animal and Veterinary Sciences (AL4AnimalS), Portugal Department of Veterinary Science, University Lusófona, 1300-477 Lisbon, Portugal; 5CITAB—Centre for the Research and Technology of Agro-Environmental and Biological Sciences, University of Trás-os-Montes and Alto Douro, 5000-801 Vila Real, Portugal; 6Department of Anesthesiology and Pain Management, Facultad de Ciencias Veterinarias, Universidad de Buenos Aires, Buenos Aires C1427CWN CABA, Argentina

**Keywords:** cat, ovariectomy, quadratus lumborum block, regional analgesia, sacrococcygeal epidural

## Abstract

**Simple Summary:**

Regional anaesthesia techniques have been employed for neutering cats, offering effective pain relief. However, there is limited research on the utilisation of an epidural sacrococcygeal (ScE) and ultrasound-guided quadratus lumborum block (QLB) in feline subjects. This study aims to compare the effects of QLB and ScE, both administered with 0.25% bupivacaine, in cats undergoing ovariectomies. The evaluation included the intraoperative heart rate, respiratory rate, and systolic and mean blood pressure as well as opioid consumption for intraoperative rescue analgesia. Postoperatively, a Feline Grimace scale and assessment of motor blockade were employed. The findings of this study suggest that QLB may represent a viable alternative to ScE for perioperative pain management in cats undergoing elective ovariectomy. The QLB exhibited several advantages over ScE, such as a lower incidence of hypotension, a shorter extubation time, and reduced postoperative motor block.

**Abstract:**

Background: Ultrasound-guided quadratus lumborum block (QLB) and sacrococcygeal epidural anaesthesia (ScE) have been used for neutering cats, providing effective pain relief. Objectives: To compare the effects of the QLB with those of ScE in cats undergoing ovariectomies. Methods: Feral cats undergoing ovariectomy were premedicated with dexmedetomidine (20 μg kg^−1^) and methadone (0.2 mg kg^−1^) intramuscularly. Anaesthesia was induced with 2–4 mg kg^−1^ of propofol intravenously and maintained with isoflurane in oxygen. The cats were randomly allocated to the groups QLB (bilateral QLB with 0.4 mL kg^−1^ of 0.25% bupivacaine) and ScE (0.3 mL kg^−1^ of 0.25% bupivacaine). Hemodynamic data and analgesia rescue were collected at four intraoperative periods. The pain scale and motor block were assessed in both groups during the postoperative period. Results: The ScE results in increased hypotension, prolonged extubation time, and higher postoperative motor block than the QLB (*p* < 0.05). The QLB and ScE groups required a similar number of intraoperative rescues and presented the same postoperative pain scale classification. Conclusions: The QLB with 0.25% bupivacaine is a potential alternative to ScE with 0.25% bupivacaine in perioperative pain management in elective cat ovariectomy. The QLB promoted less hypotension and postoperative motor block when compared with the ScE group.

## 1. Introduction

The ovariectomy is commonly used in veterinary medicine as a model of acute intraoperative abdominal pain [1,2,3]. Maintaining the stability of the autonomic nervous system is crucial during general anaesthesia, underscoring the need for adequate intraoperative analgesia [4,5]. Regional anaesthesia is widely accepted as an optimal method to prevent perioperative pain [3,6], as it effectively blocks nociceptive transmission [3,7].

Different regional anaesthesia techniques have been used for neutering cats, providing effective pain relief [3,7,8,9]. One such technique is ultrasound-guided quadratus lumborum block (QLB), which has been described in cadavers [10] and found to be effective in cat ovariectomies [9]. The QLB involves injecting a local anaesthetic solution between the quadratus lumborum and psoas muscles [11,12]. It has been demonstrated in feline cadavers that the injection of dye between the quadratus lumborum and psoas muscles at the level of the second lumbar vertebra (L2) successfully stains the last thoracic and first spinal nerves and the sympathetic trunk [10]. Moreover, ultrasound-guided QLB has been shown to reduce opioid consumption and provide perioperative analgesia in dog ovariectomies [12,13].

Conversely, employing a sacrococcygeal approach for epidural anaesthesia is widely acknowledged as the suitable choice for perioperative analgesia in cats [14], and it is characterised by its minimal complication rate [8,15,16]. In addition, studies have reported the analgesic effects of epidural anaesthesia in cats undergoing perineal or hind limb surgeries [5] and ovariectomies [8]. However, there is currently a lack of studies comparing the efficacy and outcomes of QLB and epidural anaesthesia in cats.

This study aims to compare the effects of QLB with those of sacrococcygeal epidural anaesthesia (ScE) in cats undergoing ovariectomies. For this purpose, we tested a null hypothesis that QLB with bupivacaine is non-inferior to a standard sacrococcygeal epidural anaesthesia and an analgesia technique in cats undergoing ovariectomies under general anaesthesia. The hypothesis is that ScE will provide comparable perioperative analgesia to QLB, while QLB will offer better cardiovascular stability, faster recovery, and reduced motor block.

## 2. Materials and Methods

### 2.1. Animals

The protocol of this study was approved by the Institutional Animal Care and Use Committee of Lusófona University (no. 15/2022). Fifty-eight adult cats admitted for neutering were included in this clinical prospective, randomised, blinded study between November and December of 2022. Responsible person informed consent was obtained for all the animals enrolled. This report follows the Consolidated Standard of Reporting Trials (CONSORT) guidelines.

We conducted an a priori sample size estimation using G*Power 3.1.9.7. We determined a minimum sample size of 50 cats, considering an estimated effect size of 0.4 (Appendix A) for the incidence of the hypotension variable. This calculation utilised an alpha error of 0.05 and a power of 0.8. A total of 58 feral female cats were enrolled and randomised to receive an ultrasound-guided QLB or ScE approach as part of the anaesthesia protocol.

Before anaesthesia induction, all the cats underwent serum biochemistry, blood haematology, and a physical examination under sedation. The eligibility criteria included being healthy (determined through physical examination and laboratory analysis), while pregnant females, those less than 6 months old, and animals with clinical signs of systemic disease were excluded. Animals showing pain during the pre-anaesthetic assessment were excluded from the study. The cats were transported to the hospital in individual cages, following cat-friendly guidelines for handling and manipulation. As per the guidelines, water was removed during transport, and food was withheld for at least 6 h prior to surgery. Upon arrival, the cats were assigned numbers according to their admission order, and their weights and body condition scores (BCS) on a scale from 1 to 9 were recorded. 

### 2.2. Anaesthesia Protocol

The cats received an intramuscular injection of dexmedetomidine (20 μg kg^−1^; Dexdomitor 0.5 mg mL^−1^, Orion, Finland) and methadone (0.2 mg kg^−1^; Semfortan 10 mg mL^−1^, Dechra, Italy) for premedication into the epaxial musculature. Under sedation and with aseptic conditions, a cephalic vein was catheterised to provide medications and fluids at a rate of 3–5 mL kg^−1^ (RingerVet B. Braun Medical Inc., Melsungen, Germany) during the perioperative period. Hair was clipped from the sternum to the pre-lumbar regions and from the cranial portion of the iliac crest to the third coccygeal vertebra to ensure blinding of the study. The skin was prepared aseptically, and a heating blanket (Thermal Blanket Carbonvet cage, B. Braun Medical Inc., Shanghai, China) was used to prevent hypothermia. After adequate sedation, two minutes of preoxygenation was provided. General anaesthesia was induced by intravenous (IV) propofol injection (Propofol 10 mg mL^−1^; B. Braun Medical Inc., Germany) to effect 12–15 min after premedication. Subsequently, tracheal intubation was performed using a cuffed endotracheal tube with a size of 3.5 to 4.5 mm, preceded by the application of 0.1 mL of 2% lidocaine (Lidocaine 20 mg mL^−1^, B. Braun Medical Inc., Melsungen, Germany) to the larynx. Spontaneous breathing was maintained throughout the procedure using a non-rebreathing system with a 100% inspired oxygen concentration and a fresh gas flow rate of 250 mL kg^−1^ min^−1^. Isoflurane (IsoFlo; Zoetis, Madrid, Spain) was administered to maintain an appropriate depth of anaesthesia, with an initial target vaporiser setting of 0.8 ± 0.2%. Monitoring was initiated immediately after premedication using a BeneVision N15 monitor (Mindray, Shenzhen, China). Continuous monitoring included an assessment of the heart rate (HR; beats per minute) and rhythm, respiratory rate (RR; breaths per minute), end-tidal carbon dioxide (Pe’CO_2_; mmHg), oesophageal temperature (°C), oxygen haemoglobin saturation (SpO_2_; %), and end-tidal isoflurane concentration (%). Non-invasive blood pressure measurements, including systolic (SAP) and mean arterial pressure (MAP), were taken at 3-minute intervals using an oscillometric method with a number 2 cuff. Hypotension was classified as SAP < 90 mmHg or MAP < 60 mmHg. The Fe’Iso was decreased by 20% during episodes of hypotension when the anaesthetic plan ensured the absence of the palpebral reflex, jaw tone, and purposeful movements. If reducing isoflurane was not effective in managing the hypotension, a bolus infusion of lactated Ringer’s solution at a rate of 5 mL kg^−1^ over 10 min was administered. When required, ephedrine was administered IV (0.1 mg kg^−1^; Labesfal). The anaesthesia chart was completed by an anaesthesiology professor (LR) who was blinded to the groups and made all the decisions deemed important for the animal during the procedure.

### 2.3. Block Techniques

The animals were randomly allocated to two groups using a random sequence generator from random.org. The cats in the QLB group received 0.4 mL kg^−1^ of 0.25% bupivacaine (Bupivacaine 2.5 mg mL^−1^; B. Braun Medical Inc., Germany) per hemiabdomen. The cats in the ScE group received 0.3 mL kg^−1^ of 0.25% bupivacaine (Figure 1). Both regional techniques were performed by the same anaesthetist (JDS). The group assignment was only disclosed to the attending anaesthetist and the assisting personnel.

Under general anaesthesia, the ultrasound-guided QLB was performed on cats positioned in lateral recumbency, with the side to be injected facing upward, using a portable ultrasound machine (TE7; Mindray, Shenzhen, China) equipped with a 3 to 13 MHz linear array probe (L12-4s; Mindray, Shenzhen, China). The skin was prepared with alcohol, and the ultrasound transducer was placed caudal to the last rib, perpendicular to the spine, with its mark facing dorsally. The correct position was confirmed by visualising the transverse process of L2, the quadratus lumborum, the psoas minor, and the abdominal wall muscles. Then, using an in-plane technique, a 22-gauge, 55 mm nerve block needle (Lococare; Belphar, Portugal) attached to a 2 mL syringe was advanced ventrodorsally through the lateral aspect of the abdominal wall [10]. Once the needle tip reached the interfascial plane between the quadratus lumborum and psoas minor muscles, the calculated volume was injected. After completing the injections of the first target site, the animal was repositioned to inject the contralateral hemiabdomen. Visualisation of the ultrasound hydrodissection of the interfascial plane confirmed the correct administration. The procedure was repeated in the contralateral hemiabdomen.

The cats in the ScE group were positioned in sternal recumbency, with the pelvic limbs pulled forward and the tail hanging. Under aseptic conditions, the sacrococcygeal space was identified. A 22-gauge, 55 mm insulated needle (Lococare; Belphar, Portugal) with an extension line prefilled with 0.25% bupivacaine was used, connected to a peripheral nerve stimulator (NS). The positive lead of the NS was placed on the skin over the caudal aspect of the right thigh. The NS was set to a fixed electrical current of 0.7 mA, a frequency of 2 Hz, and a pulse of 0.1 ms [17]. The needle was inserted at a 30° angle toward the tail to reach the intervertebral junction between the last sacral and first coccygeal vertebrae. The needle was advanced gently until the lateral movement of the tail was detected. The epidural injection was confirmed through the colour flow Doppler test (CFDT) performed at the lumbosacral space [18], administering 0.2 mL of saline. The cats that exhibited a motor response of the tail and a positive CFDT were then administered 0.3 mL kg^−1^ of 0.25% bupivacaine over 45 s.

### 2.4. Data Collection 

During the anaesthesia, the HR, RR, SAP, and MAP were continuously monitored. These parameters were evaluated and recorded at five time points: baseline values before skin incision (t0), at the time of skin incision (t1), at the first ovary removal (t2), at the second ovary removal (t3), and skin closure (t4). Approximately 10 min after administering the assigned block, an experienced surgeon (JM, PC) initiated the ovariectomy procedure. In the cases where there was an increase in the HR exceeding 20%, coupled with an increase in MAP and RR compared to the t0 values, accompanied by the absence of a palpebral reflex, mandibular tone, or purposeful movements in the cats, an intravenous bolus of fentanyl (2 μg kg^−1^; Fentadon; 50 µg mL^−1^, Dechra, Italy) was administered as rescue analgesia, considering it as a nociceptive stimulation requiring intervention.

### 2.5. Postoperative Monitoring

The time from turning off the Isoflurane until the presence of a jaw tone or swallowing reflex was recorded as the extubation time. Following extubation, once the cats were evaluated and deemed stable based on parameters such as HR, RR, oral mucous membrane colour, and temperature, they were administered subcutaneous meloxicam (0.2 mg kg^−1^; Metacam 2 mg mL^−1^, Boehringer Ingelheim Vetmedica Gmbh, Ingelheim, Germany) for postoperative pain management. Subsequently, the cats were then transferred to clean and individual cages to facilitate their recovery.

The real-time Feline Grimace Scale was utilised two hours after extubation to assess the pain scores and was always performed by the same trained anaesthetist (LR). The cats with a pain score higher than four in ten were given an intramuscular injection of methadone (0.2 mg kg^−1^; Semfortan 10 mg mL^−1^, Dechra, Italy). 

During the pain evaluation, photographs were taken and later reviewed independently by four trained blinded evaluators (MG, JM, PC, SAP). Each evaluator was assigned a Feline Grimace Scale score based on their pain assessment. These evaluations were then used to compare the effectiveness of the different approaches in providing postoperative analgesia.

Descriptive rating scales from Haro et al., 2016 (Table 1) were used to evaluate the residual motor blockade [19]. Motor evaluations were performed 2 and 6 h after extubation to assess the extent of motor impairment. 

### 2.6. Statistical Analysis

Statistical analysis was performed using SPSS version 26 (IBM SPSS Statistic; IBM Corp., New York, NY, USA). The data were tested for normal distribution using the Shapiro–Wilk test. Non-normally distributed variable BCS and Grimace Scale Scores were presented as the median and interquartile range (IQR), with median comparisons analysed using the Mann–Whitney U test. For the normally distributed variables, such as demographic data, blood test results, anaesthesia time, and extubation duration, the mean and standard deviation were provided. Furthermore, means comparisons between the groups were performed using the t-test. In the rescue analgesia, the hypotension and motor block variables χ2 test was used. For inter-observer concordance, Kendall’s coefficient of concordance (Kendall’s W) was performed. The significance was set at *p* < 0.05.

## 3. Results

A total of 58 female Domestic Short Hair cats scheduled for ovariectomy were initially enrolled in the study. Eight cats were excluded from the analysis: two were under six months of age, and six had systemic diseases. A pre-anaesthetic examination showed that the cats were not in pain. Therefore, the final study population consisted of 50 cats with 25 cats per group (Figure 1).

Table 2 presents the patient characteristics, including the body weight, BCS, blood test results, and anaesthesia and extubation times. It was observed that the animals in the ScE group had a longer extubation time compared to those in the QLB group [8.40 vs. 11.40 min, respectively (*p* = 0.002)].

During the ultrasound scanning for the QLB, landmarks were successfully identified, and hydrodissection in the target interfascial plane was observed in all the injections. In the ScE group, tail myoclonus in response to NS and a positive CFDT were registered in all the cats before the bupivacaine injection. No complications related to the technique were observed in either group.

Figure 2 provides a summary of the hemodynamic and respiratory variables assessed during the intraoperative period. No significant differences were observed in the mean HR between the ScE and QLB groups at all the evaluated time points. However, the mean RR was significantly lower in the ScE group compared to the QLB group at t0 (*p* = 0.050), t1 (*p* = 0.026), and t3 (*p* = 0.014) (Figure 2).

Regarding blood pressure, the ScE group exhibited significantly lower mean values in the SAP and MAP compared to the QLB group at t0 (*p* = 0.014; *p* = 0.005), t1 (*p* = 0.005; *p* = 0.001), and t2 (*p* = 0.004; *p* = 0.002) (Figure 2). At t3, the SAP mean was significantly lower in the ScE group compared to the QL group (*p* = 0.037) (Figure 2). There was a statistically significant difference in hypotension occurrence, which was detected in 11/25 (44%) and 4/25 (16%) in the ScE and QLB groups, respectively (*p* = 0.031) (Table 3). 

The incidence of rescue analgesia was similar between the groups (*p* = 0.351), with 3/25 (12%) and 5/25 (20%) in groups ScE and QLB, respectively (Table 3).

In all the cats, the pre-surgical pain score was zero. The “real-time” Feline Grimace Scale score was below four for all the patients, and no rescue analgesia was required during the recovery period. The Feline Grimace Scale score showed no significant difference between the SCE and QLB groups (*p* = 0.877), with median values of 1 and 2, respectively (with strong agreement (Kendall W = 0.733) among the four evaluators). 

In the motor block evaluation, a statistically significant difference between the groups was observed at the 2-hour mark after extubation (*p* < 0.05) (Table 4). However, by the 6-hour evaluation and before discharge, normal muscle function had returned in all patients from both groups. No unexpected neurological complications were observed in any of the animals.

## 4. Discussion

This study indicates that sacrococcygeal epidural injection and QLB with 0.25% bupivacaine offer comparable perioperative analgesia for feral cats undergoing elective ovariectomy. However, the QLB group demonstrated advantages, such as a faster extubation time and less postoperative motor block, compared with the ScE group. Additionally, the QLB group exhibited a reduced incidence of hypotension compared to the ScE group.

The use of ultrasound-guided QLB performed between the quadratus lumborum muscle and the psoas minor has been shown to affect the ventral branches of the last thoracic and first lumbar spinal nerves, along with compromise of the sympathetic nerves in cats [10]. On the other hand, the ScE approach has been found to provide epidural distribution and perioperative analgesia in caudal pelvic surgeries [5] and ovariohysterectomies in cats [8]. The observed improvement in perioperative analgesia, as indicated by the number of rescue interventions and pain scales, in both the QLB and ScE groups, is consistent with other clinical studies conducted in neutered cats [8,9]. Our findings support that adding QLB or ScE to a multimodal analgesic protocol might provide similar perioperative analgesia in cats undergoing ovariectomy.

Regarding HR, no significant differences were observed between the QLB and ScE groups. However, the mean RR was significantly lower in the ScE group at t0, t1, and t3 compared to the QLB group. These findings suggest that administering 0.3 mL kg^−1^ of bupivacaine in the ScE group may have led to respiratory compromise and diaphragmatic dysfunction, similar to what has been described in dogs [20,21].

The observed hypotension in the QLB group was lower than that reported in dogs under QLB [13]. Additionally, the ScE group exhibited lower arterial blood pressure than the QLB group. The higher occurrence of hypotension in the ScE group could be attributed to the more extensive sympathetic block and subsequent vasodilation associated with epidural administration of bupivacaine, as described in previous studies involving cats [5,22]. The lower prevalence of hypotension in the QLB group supports that this block may result in less hypotension than ScE in cats undergoing ovariectomy.

In this study, the QLB successfully prevented the need for rescue analgesia due to nociception in 80% of the cats. These results are comparable to the findings of a previous study on ovariectomy in bitches using QLB, which reported a success rate of 68.8% without the need for rescue analgesia [13]. However, in terms of nociceptive prevention, the ScE group in this study had a higher success rate (88%) compared to other studies that utilised epidural administration for cats undergoing ovariohysterectomies (60%) [8] and a similar success rate to a dog study (81.8%) [13]. Our results may be explained by the association of the use of the NS and CFDT. In addition, the number of intraoperative rescue analgesics required was similar between the two groups, with no significant difference observed. These findings suggest that ultrasound-guided QLB has comparable intraoperative analgesic effects to ScE in cats undergoing ovariectomy procedures.

The longer extubation time observed in the ScE group could be explained by the direct reduction in spinal descending excitatory modulation of afferent input [23], suggesting that ScE induces more prolonged recovery time in the postoperative period [8,24].

Pain assessment in the postoperative period is important for comparing analgesic techniques. The objective was to identify an effective intraoperative analgesic strategy that promotes an excellent postoperative period. The Feline Grimace Scale has been validated to assess postoperative pain in cats [25,26,27]. The median values obtained from the Feline Grimace Scale were consistently low in both the QLB and ScE groups. These results indicate that QLB and ScE may effectively control analgesia during the immediate postoperative period without significant differences. ScE has been effective in the postoperative period in cats undergoing neutered surgery [8]. However, to our knowledge, no studies have evaluated the analgesic efficacy of the QLB during the postoperative period for this type of surgery in cats. 

The ScE provided a more effective motor block within the first 2 h compared to the QLB group. Epidural administration of bupivacaine has been associated with sensory and motor block lasting up to four hours after administration [28,29]. The results obtained with 0.25% bupivacaine in the ScE group are consistent with the literature, as no motor block was observed during the 6-hour evaluation. In contrast, no motor block was observed in the QLB group, which is in accordance with the results of previous studies in the veterinary field that did not report motor block [12,13]. These findings indicate that QLB may reduce motor block during the first six hours of the postoperative period compared to ScE.

This study has several limitations that should be acknowledged. Firstly, the lack of information on the inspiratory and expiratory fraction of isoflurane hinders the comparison of anaesthetic requirements during the intraoperative period. Secondly, the block techniques were performed by a single operator, which helps reduce bias but may limit the results’ generalisability to cases involving multiple operators. The time elapsed between the surgical procedure and block execution may influence the results, given that bupivacaine was used. Furthermore, the bupivacaine concentration used for the ScE could have been reduced to minimise motor block, as Abelson et al. (2011) suggested [30]. However, further studies are needed to evaluate this possibility. Additionally, the specific concentration and volume of the blocks utilised in this study may have influenced the results, highlighting the need for future research to investigate the effects of different dosages. Lastly, it is essential to note that the use of the Feline Grimace Scale in feral cats is not yet validated, and its impact on the scoring of animals in this study remains unknown.

## 5. Conclusions

This study suggests that ultrasound-guided QLB using 0.4 mL kg^−1^ of 0.25% bupivacaine may be a suitable alternative to ScE with 0.3 mL kg^−1^ of 0.25% bupivacaine for perioperative pain management in cats undergoing elective ovariectomy. The QLB demonstrated advantages over the ScE, including a lower incidence of hypotension, a shorter extubation time, and reduced postoperative motor block. However, further research is needed to determine the optimal dosage of local anaesthetic, evaluate the overall success rate of these techniques, and assess the associated risks.

## Figures and Tables

**Figure 1 vetsci-11-00025-f001:**
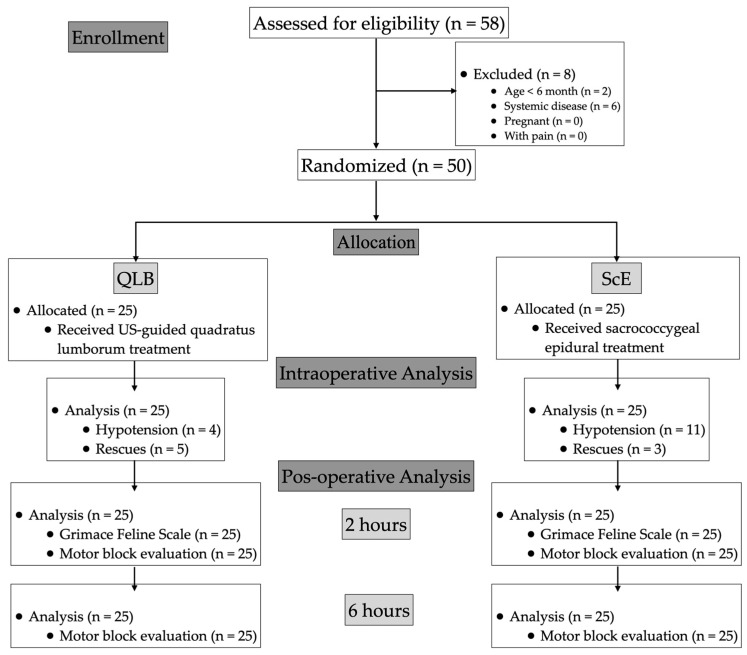
Consolidated standards of reporting trials (CONSORT) flowchart describing patient progress through the study. QLB: ultrasound-guided quadratus lumborum block with bupivacaine 0.25%; ScE: sacrococcygeal epidural anaesthesia with bupivacaine 0.25%.

**Figure 2 vetsci-11-00025-f002:**
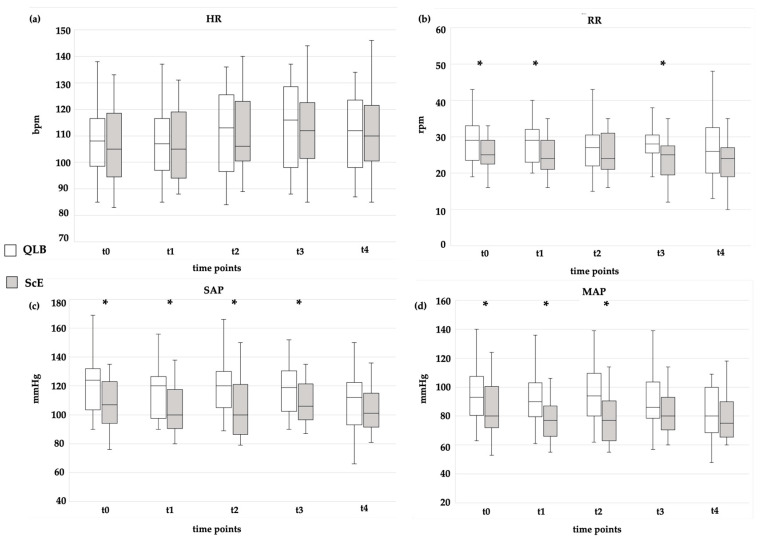
Intraoperative in (**a**) mean heart rate (HR), (**b**) mean respiratory rate (RR), (**c**) mean systolic arterial pressure (SAP), and (**d**) mean of the mean arterial pressure (MAP) in two treatment groups in five different surgical times. bpm: beats per minute; mmHg: millimetres of mercury; QLB: ultrasound-guided quadratus lumborum block with bupivacaine 0.25%; rpm: breaths per minute; ScE: sacrococcygeal epidural anaesthesia with bupivacaine 0.25%. Differences between groups at the same timepoint were analysed, and significant results (*p* < 0.05) were represented with *.

**Table 1 vetsci-11-00025-t001:** Descriptors of motor function scores, adapted from Haro et al., 2016.

Rating	Classification	Description
0	Normal motor response	Normal ability to walk or stand
1	Partial motor blockade	Incomplete ability to bear weight, incomplete flexion of the stifle, muscle tone weaker than in the contralateral limb
2	Complete motor blockade	Inability to stand and walk with the blocked limb, knee in extension touching the floor, limb paralysed

**Table 2 vetsci-11-00025-t002:** Means and standard deviations of demographic data, blood test results, anaesthesia time, surgical time, and time to extubation in cats undergoing ovariectomy. Body condition scores (BCS) are presented as median and interquartile range [IQR]. Significance was set at *p* < 0.05 (*). QLB: ultrasound-guided quadratus lumborum block group with bupivacaine 0.25%; ScE: sacrococcygeal epidural with bupivacaine 0.25%.

Variable	QLB (n = 25)	ScE (n = 25)	*p* Value
Body weight (kg)	3.50 ± 0.93	3.50 ± 0.68	0.993
BCS (1–9) (median [IQR])	4 [2]	4 [2]	0.968
Haematocrit (%)	33.00 ± 0.76	34.78 ± 1.33	0.325
Glucose	220.36 ± 15.86	209.24 ± 17.79	0.572
Total protein (g dL^−1^)	7.55 ± 0.14	7.48 ± 0.12	0.420
Platelets (×10^3^ µL^−1^)	255.68 ± 24.60	285.76 ± 27.64	0.682
Anaesthesia time (min)	27.72 ± 3.36	28.12 ± 3.89	0.699
Time to extubation (min)	8.40 ± 2.89	11.40 ± 3.50	0.002 *

**Table 3 vetsci-11-00025-t003:** Percentage of rescue analgesia required intraoperatively per group and percentage of hypotension detected per group. Statistical differences were set at *p* < 0.05 (*). QLB: ultrasound-guided quadratus lumborum block group with bupivacaine 0.25%; ScE: sacrococcygeal epidural with bupivacaine 0.25%.

	n	QLB	ScE	*p* Value
Rescue	8/25	5/25 (20%)	3/25 (12%)	0.351
Hypotension	15/25	4/25 (16%)	11/25 (44%)	0.031 *

**Table 4 vetsci-11-00025-t004:** Postoperative classification in terms of motor block two and six hours after extubation. Statistical differences are marked with ** (*p* < 0.05). QLB: ultrasound-guided quadratus lumborum block group with bupivacaine 0.25%; ScE: sacrococcygeal epidural with bupivacaine 0.25%.

	Rating	n	QLB	ScE
Motor block				
2 h	0	29	25 (100%)	4 (16%) **
	1	7	0	7 (28%) **
	2	14	0	14 (56%) **
6 h	0	50	25 (100%)	25 (100%)
	1	0	0	0
	2	0	0	0

## Data Availability

Data are contained within the article.

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
