# Peer review of "Comparison between Bilateral Ultrasound-Guided Quadratus Lumborum Block and Sacrococcygeal Epidural in Cats Undergoing Ovariectomy"

_vetsci, 2024, doi:10.3390/vetsci11010025_

Round 1
Reviewer 1 Report
Comments and Suggestions for Authors
Thank your for this interesting and enjoyable read.
Below, you can find some suggestions.
Line 39 than instead of then
line 43 compared
Line 56 described
Authors may consider make the English more fluent at this part of the manuscript.
Which was the nociception response considered for sample size calculation. I don’t know what 0.4 effect size refers to.
How didyou evaluate pain signs preoperatively? Which signs of pain in feral, not acclimatized feral cats?
Doppler test was used during epidural administration. How? Please, described it briefly. May you included some discussion about succes rate of this verification method for epidural administration in this specie.
To my knowledge, you performed comparison between groups by using ANOva, chi squared..but you did not perform superiority or inferiority test based on AUC analysis so be cautions when discussing your results regarding ‘ this was SUPERIOR to..’. Please review throught the manuscript.
Line 306 this affirmation can be confusing considering that you did not evaluate or consider postop sedation during pain evaluation. Please, include or discuss it,
Lines 309-310 and 340 are contradictory. Please, correct it.
Line 344. Consider use suitable instead of viable
Comments on the Quality of English Language
See previous comments
Author Response
Many thanks for your e-mail concerning our article. We have improved our work to attend to all the reviewer comments. All changes in the manuscript are highlighted in yellow.
R: Line 39 than instead of then
AR: Thank you for your comment. We improved the sentence as suggested.
R: line 43 compared
AR: Thank you for your comment. We improved the sentence as suggested.
R: Line 56 described
AR: Thank you for your comment. We improved the sentence as suggested.
R: Authors may consider make the English more fluent at this part of the manuscript.
AR: Thank you for your comment. In this version, the English was reviewed by a native English speaker.
R: Which was the nociception response considered for sample size calculation. I don’t know what 0.4 effect size refers to.
AR: Thank you for your comment. We agree with your assessment. The description in our power analysis materials and methods was unclear and lacked precision. Our sample size calculation was based on the hypotension variable. We have revised the wording in this section in the manuscript in order to enhance clarity. The effect size was computed by determining the expected difference between the two groups (QLB and ScE) and then dividing it by the expected standard deviation of one of the groups. According to prior studies, we anticipated the effect size to be 0.4.
R: How did you evaluate pain signs preoperatively? Which signs of pain in feral, not acclimatized feral cats?
AR: Thank you for your comment. Only a subjective visual scale was used, noting signs of lameness, wounds, and changes in posture.
R: Doppler test was used during epidural administration. How? Please, described it briefly. May you included some discussion about success rate of this verification method for epidural administration in this specie.
AR: Thank you for your comment. We added the information in discussion as suggested.
R: To my knowledge, you performed comparison between groups by using ANOva, chi squared..but you did not perform superiority or inferiority test based on AUC analysis so be cautions when discussing your results regarding ‘ this was SUPERIOR to..’. Please review throught the manuscript.
AR: Thank you for your comment. You are right; we did not perform an AUC analysis. Additionally, the terms 'superior' and 'inferior' have been substituted with more appropriate terminology throughout the entire manuscript.
R: Line 306 this affirmation can be confusing considering that you did not evaluate or consider postop sedation during pain evaluation. Please, include or discuss it,
AR: Thank you for your comment. We change sedation for recovery time.
R: Lines 309-310 and 340 are contradictory. Please, correct it.
AR: Thank you for your comment. The Feline grimace scale is validated in docile domestic cats. We do not know if this scale will have the same interpretation in wild animals, hence it is listed in the limitations.
R: Line 344. Consider use suitable instead of viable
AR: Thank you for your comment. We improved the sentence as suggested.

Reviewer 2 Report
Comments and Suggestions for Authors
Dear authors thank you for submitting this interesting and well-written manuscript. Please find below some comments that I believe they will contribute to the improvement of the manuscript.
It is very strict to mention that epidural anaesthesia is the golden standard in OHE in cats. I agree that it is very helpful, but golden standard?..
How many minutes after the local anaesthesia (in both groups) did the operation started? Since anaesthesia lasted for a mean of 27-28 minutes, how would you be so sure that local anaesthesia was effective intraoperatively, knowing the pharmacokinetics of bupivacaine?
Furthermore, how would you know that the lower RR in ScE group was a results of respiratory depression? Why not the increased RR be a nociceptive response from the not-so-effective QLB?
Statistics: As far as I can understand, the design is a repeated measurements block of two groups. So, a repeated-measures ANOVA should be used. Is that the case?
I cannot find Figure 3 in your manuscript.
I cannot find the inter-observer analysis.
I believe that the presentation of figures (MAP, SAP, HR, fR) would be more informative to the readers than the box-and-whiskers graphs.
Author Response
Reviewer 2
Many thanks for your e-mail concerning our article. We have improved our work to attend to all the reviewer comments. All changes in the manuscript are highlighted in yellow.
R: It is very strict to mention that epidural anaesthesia is the golden standard in OHE in cats. I agree that it is very helpful, but golden standard?..
AR: Thank you for your comment. We improved the sentence and change for suitable choice
R: How many minutes after the local anaesthesia (in both groups) did the operation started? Since anaesthesia lasted for a mean of 27-28 minutes, how would you be so sure that local anaesthesia was effective intraoperatively, knowing the pharmacokinetics of bupivacaine?
AR: Thank you for your comment. Effectively, the initiation of the procedure occurred approximately 10 min after the administration of the blocks (data collection area) Bupivacaine was used for greater benefit in terms of postoperative analgesia duration. This information was included in the limitations.
R: Furthermore, how would you know that the lower RR in ScE group was a results of respiratory depression? Why not the increased RR be a nociceptive response from the not-so-effective QLB?
AR: Thank you for your comment. We agree that there could be several variables causing respiratory depression or a reduction in RR in the ScE group. However, ScE could be one of the causes, as mentioned by Lebeaux, M.I. Experimental epidural anaesthesia in the dog with lignocaine and bupivacaine. Br. J. Anaesth. 1973, 45, 549-55. DOI: 10.1093/bja/45.6.549 and Castro, D.S.; Soares, J.H.; Gress, M.A.; Otero, P.E.; Marostica, E.; Ascoli, F.O. Hypoventilation exacerbates the cardiovascular depression caused by a high volume of lumbosacral epidural bupivacaine in two isoflurane-anesthetized dogs. Vet. Aneasth. Analg. 2016, 43, 235-7. DOI: 10.1111/vaa.12320
R: Statistics: As far as I can understand, the design is a repeated measurements block of two groups. So, a repeated-measures ANOVA should be used. Is that the case?
AR: Thank you for your comment. The repeated measures ANOVA is typically employed when assessing the same measure across more than two time points. For cases involving only two time points, a paired t-test is generally adequate. In our study, we measured the same variable at five time points for both groups (QLB and ScE). However, our experimental design was specifically geared towards identifying group differences within each individual time point rather than between them. Therefore, we did not intend to compare variables across different time points (T0, T1, T2, T3, and T4). Consequently, since our focus lies in comparing variables between the two groups at singular time point rather than across multiple time points, employing an ANOVA t-test suffices for our analysis.
R: I cannot find Figure 3 in your manuscript.
AR: Thank you for your comment. We removed the inter-operator post-op pain assessment and Figure 3 form the study
R: I cannot find the inter-observer analysis.
AR: Thank you for your comment. For inter-observer analysis was used Kendall's coefficient of concordance (Kendall´s W). Was mentioned in statistics analysis and in results line 204-206.
R: I believe that the presentation of figures (MAP, SAP, HR, fR) would be more informative to the readers than the box-and-whiskers graphs.
AR: Thank you for your comment. We improved the figure.
